# Outage Performance of Interference Cancellation-Aided Two-Way Relaying Cognitive Network with Primary TAS/SC Communication and Secondary Partial Relay Selection

**Pham Minh Nam [1], Ha Duy Hung [2],\* [ID], Lam-Thanh Tu [3], Pham Viet Tuan [4] [ID], Tran Trung Duy [5] [ID] and Tan Hanh [5]**

[1]  Faculty of Electronics Technology, Industrial University of Ho Chi Minh City,
    Ho Chi Minh City 700000, Vietnam
[2]  Wireless Communications Research Group, Faculty of Electrical and Electronics Engineering,
    Ton Duc Thang University, Ho Chi Minh City 700000, Vietnam
[3]  Communication and Signal Processing Research Group, Faculty of Electrical and Electronics Engineering,
    Ton Duc Thang University, Ho Chi Minh City 700000, Vietnam
[4]  Faculty of Physics, Universityof Education, Hue University, Thua Thien Hue 530000, Vietnam
[5]  Posts and Telecommunications Institute of Technology, Ho Chi Minh City 700000, Vietnam
\*  Correspondence: haduyhung@tdtu.edu.vn

**Abstract:** In this paper, we propose a two-way relaying scheme using digital network coding in an underlay cognitive radio network. In the proposed scheme, the transmit antenna selection and selection techniques are combined using a primary transmitter and a primary receiver, respectively. In the secondary network, two source nodes that cannot directly communicate attempt to exchange their data with each other. As a result, the relaying technique using partial relay selection is applied to assist the data exchange. Particularly, at the first time slot, the selected secondary relay applies an interference cancellation technique to decode the data received from the secondary sources. Then, the selected relay uses digital network coding to send XOR-ed data to the sources at the second time slot. We first derive the outage probability of the primary network over block the Rayleigh fading channel. Then, the transmit power of the secondary transmitters including the source and relay nodes are calculated to guarantee the quality of service of the primary network. Finally, the exact closed-form formulas of the outage probability of the secondary sources over the block Rayleigh fading channel are derived, and then verified by computer simulations using the Monte Carlo method.

**Keywords:** interference cancellation; two-way relaying network; digital network coding; underlay cognitive radio; outage probability; partial relay selection

## 1. Introduction

In recent years, cognitive radio networks (CRNs) [1] have become an attractive topic that has gained significant attention among researchers. Underlay spectrum sharing (or underlay CRN) is one efficient technique that allows secondary users (SUs) to access bands licensed to primary users (PUs). To satisfy a pre-determined interference constraint, SUs must adjust the transmit power, using the instantaneous channel state information (CSI) of the interference links from SUs to PUs [2–6]. However, it is difficult to implement spectrum-sharing methods in [2–6] due to the requirement of high synchronization between PUs and SUs. In [7–9], the transmit power of the secondary transmitters is calculated using the expected values of CSI under the constraint of a target outage probability (OP) of the primary network. Due to the limited transmit power and co-channel interference from the primary operation, the performance of the secondary network is severely degraded [9]. To improve the secondary performance, relaying schemes using intermediate relays (see [10–15]) are commonly employed. Particularly, whilst some published works [6,7,10] studied dual-hop relaying underlay CNRs, others [2,8] considered multi-hop relaying ones. To further enhance the performance of the secondary network in dual-hop relaying models, relay

selection methods were investigated in [3,16–18]. The authors in [16,17] proposed partial relay selection (PRS) methods, where the best relay was selected using the CSI of the source–relay links. In contrast to PRS, the full-relay selection (FRS) method in [18] used the CSI of the source–relay and relay–destination links to find the best relay. Therefore, the implementation of PRS is much simpler than that of FRS, and hence, in this paper, the PRS is employed to enhance the OP performance for the secondary network. However, published works [2,3,6–10,16–18] have not studied the non-orthogonal multiple access (NOMA) technique which can significantly increase the data rate for the secondary network.

In [4,5,19–25], the underlay CRNs using NOMA were reported. Applying NOMA, a secondary source can communicate with different secondary destinations at the same time and the same frequency, and the secondary destinations have to perform the interference cancellation technique (ICT) to extract their desired signals. As a result, NOMA provides a much higher data rate and multiplexing gain for the underlay CRNs, as compared with conventional orthogonal multiple access (OMA). In [19], a dual-hop underlay CRN using NOMA was introduced, where a secondary source could send its data to two secondary destinations thanks to an amplify-and-forward (AF) secondary relay. Compared to the corresponding OMA scheme, the scheme proposed in [19] obtained better performance, in terms of OP and throughput. The published works [4,5] proposed that dual-hop co-operative decode-and-forward (DF) underlay CRNs adopted NOMA. The authors in [20] evaluated the throughput of a multi-user underlay CRN using the user selection method. In addition, the published work [20] proposed an intelligent switching strategy between NOMA and OMA. In [21], the authors proposed a CSI-based power control for an adaptive NOMA/OMA switching scheme in the uplink underlay CRNs. In [22], PRS was applied for the NOMA-assisted underlay CRNs to increase the reliability of data transmission between a secondary base station and multiple secondary users. In addition, the authors in [22] also studied the impact of the imperfect SIC on pairwise error probability (PEP). In [23], the OP of a cooperative NOMA scheme with the imperfect SIC in the underlay CRNs was evaluated. Th published work [24] proposed the cognitive NOMA scenario for Internet of Things (IoTs) networks using short-packet communications under joint impact of imperfect CSI and SIC problems. The authors in [25] considered a reconfigurable intelligent surface (RIS)-aided cognitive NOMA model, where RIS (instead of the secondary relays) was deployed to serve two secondary destinations. However, the published works [4,5,19–25] were only concerned with one-way relaying underlay CRNs.

Two-way relaying (TWR) [26] is an efficient method that provides higher throughput for bidirectional communication networks. In TWR, two sources attempt to exchange their data via the assistance of one or many common relays. In conventional dual-hop TWR networks [27], the data exchange is performed via 04 orthogonal time slots, and hence the obtained data rate is 02/04 (02 data/04 time slots). In digital network coding (DNC) TWR or three-phase TWR [27–31], the sources use the first two time slots to send their data to the common relay. Then, this relay attempts to decode the received data, performs the XOR operation, and finally transmits the XOR-ed data to the sources at the third time slot. The DNC TWR approach can reduce one transmission time slot, and hence the obtained data rate is 02/03 (02 data/03 time slots). The authors in [27,28] proposed max–min relay selection approaches to reduce the bit error rate (BER) for the DNC TWR networks. In [29], the three-phase TWR model using FRS, with the presence of untrusted relays, was reported. The published works [30,31] studied the OP performance of the DNC TWR networks operating in the radio-frequency energy harvesting (RF-EH) environment. Moreover, both PRS and FRS were considered and evaluated in [31]. Analog network coding (ANC) TWR or two-phase TWR was introduced in [32], where two source nodes used the first time slot to transmit their data to the common relay. At the second time slot, the relay amplified the received signals, and then sent the amplified signal to both sources. Therefore, the ANC TWR scheme obtained the data rate of 02/02 (02 data/02 time slots). However, in the two-phase TWR, when amplifying the received signals, noises are also amplified and accumulated at the sources. Unlike ANC TWR, the relay in DNC TWR can perfectly remove

noises due to the decoding operation. However, the disadvantage of DNC TWR is that this technique must use three time slots for each data exchange. To enhance the data rate for DNC TWR, the published works [33–35] applied the interference cancellation technique (ICT) at the common relay, and hence the schemes proposed in [33–35] also obtained the data rate of 02/02.

This paper proposes the DNC TWR scheme operating in the underlay spectrum sharing mode. In the primary network, the transmit antenna selection (TAS) and selection combining (SC) techniques are used at the transmitter and receiver nodes, respectively. In the secondary network, the DNC TWR approach with PRS and ICT is used to enhance the OP performance.

### 1.1. Related Works

In contrast to [27–31], in this paper, ICT is used to reduce one transmission time slot for the DNC TWR networks. Unlike [33–35], our proposed scheme applies PRS to the secondary network. Moreover, this paper considers block fading channels, in which channel coefficients do not change during one data transmission cycle, but independently change after each cycle. Although the published works [27,36,37] were concerned with TWR in the underlay CRNs, these references did not study ICT. Moreover, Ref. [36] studied the two-phase TWR model exploiting a direct link between the sources, while Ref. [37] considered a single-relay model operating on the RF-EH environment. The authors in [38] proposed the underlay TWR scheme using RIS and full-duplex transmission. However, Ref. [38] did not study relay selection as well as ICT.

To the best of our knowledge, the published works [39,40] are the most relevant to the topic of this paper. Indeed, [39,40] both considered the DNC TWR underlay CRNs using ICT. In [39], the secondary relay node was selected to maximize the channel capacity at both sources as well as to minimize the collection time of CSI. However, the main differences between this paper and Ref. [39] are given as follows: (i) different relay selection methods were proposed in this paper and [39]; (ii) in [39], block fading channel was not considered; (iii) the secondary transmitters in [39] used the instantaneous CSI of the cross-interference links to adjust their transmission power. Then, it is worth noting that this paper was developed from our previous work [40], and the main difference between this paper and [40] can be listed as follows: (i) this paper applies PRS for the secondary network; (ii) the block fading channel was not considered in [40]; (iii) the transmit power adjustment method for the secondary transmitters in this paper is different to that in [40]; and (iv) the mathematical derivations in this paper are more challenging because we study the PRS and block fading channel.

### 1.2. Motivation and Main Contribution

The motivation and main contribution of this paper can be summarized as follows:

- We propose the TAS/SC technique for the primary network to enhance the OP performance for the primary network as well as to increase the spectrum access possibility of the secondary network.
- Based on the exact closed-form expression of the OP of the primary network, we derived the closed-form expressions of the transmit power for the secondary source and relay nodes under the condition that the minimum QoS of the primary network must be guaranteed.
- The PRS method is applied to the secondary network to enhance the OP performance at the secondary sources.
- We derive new exact closed-form formulas of OP at the source nodes over the block Rayleigh fading channel, which are checked and corrected with Monte Carlo simulations.

We organize this paper as follows: the introduction is in Section 1. The proposed system model is illustrated in Section 2. In Section 3, we analyzed the performance of the primary and secondary networks. The results and discussion are given in Section 4. Finally, the conclusions and useful recommendations are provided in Section 5.

## 2. System Model

In Figure 1, we present the system model of the proposed underlay CRN applying TAS/SC for the primary network, and the DNC TWR and PRS techniques for the secondary network. In the primary network, the $N_T$-antenna primary transmitter (PT) uses TAS to serve the $N_R$-antenna primary receiver (PR) using SC. In the secondary network, the secondary sources $SS_1$ and $SS_2$ exchange their data (denoted by $x_1$ and $x_2$, respectively) with each other thanks to $M$ secondary relays (denoted by $SR_m$, where $m = 1, 2, \ldots, M$). In addition, only one relay (denoted by $SR_b$) is selected to assist the $SS_1 - SS_2$ data exchange. Assume that $SS_1$ and $SS_2$ cannot directly communicate with each other due to the great distance between them, and all secondary nodes only have one antenna. As mentioned earlier, the data exchange is performed via two orthogonal time slots: i) $SS_1$ and $SS_2$, at the first time slot send its data to $SR_b$ which then uses ICT to decode the received data; ii) if $SR_b$ can correctly decode both $x_1$ and $x_2$, it performs the XOR operation, i.e., $x_\oplus = x_1 \oplus x_2$, and then sends the XOR-ed data $(x_\oplus)$ to $SS_1$ and $SS_2$ at the second time slot. If $SR_b$ only correctly decodes $x_1$ (or $x_2$), it will transmit $x_1$ (or $x_2$) to $SS_2$ (or $SS_1$) at the second time slot.

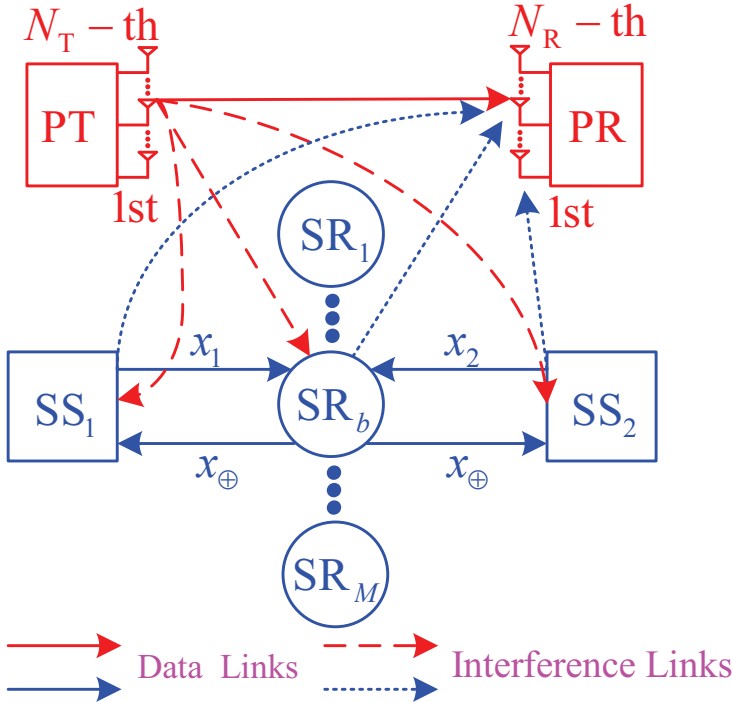

**Figure 1.** The DNC TWR model in the proposed underlay CRN.

The proposed scheme can be applied to wireless sensor networks and wireless ad hoc networks. For example, in wireless sensor networks, the secondary nodes are sensors while the primary nodes are base stations and/or mobile users in cellular mobile networks. Because of the limited size, the sensor nodes are only equipped with one antenna. Moreover, the sensor networks have to operate on the USS mode to be able to access the bands licensed to the primary nodes.

Let $h_{AB}$ denote the channel coefficient of the $A \to B$ link, where A is a transmitter and B is a receiver, i.e., $A \in \{SS_1, SS_2, SR_m, PT_u\}$, $B \in \{SS_1, SS_2, SR_m, PR_v\}$, $PT_u(u = 1, 2, \ldots, N_T)$ denotes the $u$ − th antenna of PT and $PR_v(v = 1, 2, \ldots, N_R)$ denotes the $v$ − th antenna of PR. Then, we denote $g_{AB}$ as the corresponding channel gain: $g_{AB} = |h_{AB}|^2$, and $d_{AB}$ as a link distance between A and B. We can assume that the secondary relays are close to each other, i.e., they are in one cluster. Hence, we can write that $d_{ASR_m} \stackrel{\Delta}{=} d_{ASR}$ and $d_{SR_mB} \stackrel{\Delta}{=} d_{SRB}$, $\forall A, B, m$. Moreover, we can also assume that the secondary relays are closer to $SS_1$ than to $SS_2$ (i.e., $d_{SS_1SR} \leq d_{SS_2SR}$), and $SS_1$ is closer to PR than $SS_2$ (i.e., $d_{SS_1PR} \geq d_{SS_2PR}$).

**Remark 1.** *In the case where $d_{SS_1PR} < d_{SS_2PR}$, we simply change the role of $SS_1$ by that of $SS_2$, and vice versa. Then, considering ultra-dense wireless networks [41,42] where there are a lot of nodes that are between the radio range of $SS_1$ and $SS_2$, they can be considered the potential relays. Among these relays, we focused on the M nodes which are nearer to $SS_1$ than $SS_2$.*

Because all the A-B channels are Rayleigh fading, the probability density function (PDF) and cumulative distribution function (CDF) of $g_{AB}$ can be written, respectively, as

$$f_{g_{AB}}(x) = \Omega_{AB} \exp(-\Omega_{AB}x), F_{g_{AB}}(x) = 1 - \exp(-\Omega_{AB}x), \tag{1}$$

where $\Omega_{AB} = (d_{AB})^\mu$ and $\mu$ $(2 \leq \mu \leq 8)$ denotes a path-loss exponential.

For the ease of presentation and analysis, we assume that the random variables (RVs) $g_{PT_uB}$ ($g_{APR_v}$, $g_{ASR_m}$ and $g_{SR_mB}$) are independent and identical. Therefore, we can use the following notations:

$$\Omega_{PT_uB} \triangleq \Omega_{PTB}, \ \Omega_{APR_v} \triangleq \Omega_{APR}, \ \Omega_{ASR_m} \triangleq \Omega_{ASR}, \ \Omega_{SR_mB} \triangleq \Omega_{SRB}, \ \forall u, v, m. \tag{2}$$

**Remark 2.** *This paper considers the block Rayleigh fading channel, where $g_{AB}$ remains unchanged during one data exchange cycle, but changes independently after each cycle. As a result, we have $g_{AB} \equiv g_{BA}$ for all the A and B nodes.*

Since $d_{SS_1SR} \leq d_{SS_2SR}$, we hence apply PRS for the second hop between $SS_2$ and $SR_m$ as (see [31])

$$SR_b : g_{SS_2SR_b} = \max_{m=1,2,...,M} \left( g_{SS_2SR_m} \right). \tag{3}$$

Because $SS_2$ and $SR_m$ are in radio range of one another, we can assume that the CSI of the $SR_m - SS_2$ links are available at $SS_2$. Therefore, $SS_2$ can select the best relay $SR_m$ as presented in (3).

From (3), and using CDF in (1), we obtain CDF of $g_{SS_2SR_b}$ as

$$\begin{aligned}
F_{g_{SS_2SR_b}}(x) &= \Pr\left( \max_{m=1,2,...,M} \left( g_{SS_2SR_m} \right) < x \right) = \left[ F_{g_{SS_2SR_m}}(x) \right]^M \\
&= \left[ 1 - \exp\left( -\Omega_{g_{SS_2SR}} x \right) \right]^M \\
&= 1 + \sum_{p=1}^{M} (-1)^p C_M^p \exp\left( -p\Omega_{g_{SS_2SR}} x \right),
\end{aligned} \tag{4}$$

where $C_M^p$ denotes a binomial coefficient, i.e., $C_M^p = \frac{M!}{p!(M-p)!}$.

Then, the corresponding PDF of $g_{SS_2SR_b}$ can be expressed as

$$\begin{aligned}
f_{g_{SS_2SR_b}}(x) &= M\Omega_{g_{SS_2SR}} \exp\left( -\Omega_{g_{SS_2SR}} x \right) \left[ 1 - \exp\left( -\Omega_{g_{SS_2SR}} x \right) \right]^{M-1} \\
&= \sum_{p=0}^{M-1} (-1)^p C_{M-1}^p M\Omega_{g_{SS_2SR}} \exp\left( -(p+1)\Omega_{g_{SS_2SR}} x \right).
\end{aligned} \tag{5}$$

We now describe the operation principle of the proposed scheme. At the first time slot, $SS_1$ and $SS_2$, respectively, send $x_1$ and $x_2$ to $SR_b$, and PT uses the $t$-th antenna to send its data $(x_P)$ to PR. The received signals at the $r$-th antenna of PR and at $SR_b$, under the impact of the cross interference, can be given, respectively, as

$$y_{PR_r} = \sqrt{P_{PT}} h_{PT_tPR_r} x_P + \sqrt{P_{SS_1}} h_{SS_1PR_r} x_1 + \sqrt{P_{SS_2}} h_{SS_2PR_r} x_2 + n_{PR_r}, \tag{6}$$

$$y_{\text{SR}_b} = \sqrt{P_{\text{SS}_1}} h_{\text{SS}_1\text{SR}_b} x_1 + \sqrt{P_{\text{SS}_2}} h_{\text{SS}_2\text{SR}_b} x_2 + \sqrt{P_{\text{PT}}} h_{\text{PT}_t\text{SR}_b} x_{\text{P}} + n_{\text{SR}_b}. \tag{7}$$

In (6) and (7), $P_{\text{A}}$ is the transmit power of transmitter A, where $\text{A} \in \{\text{PT}, \text{SS}_1, \text{SS}_2\}$, $n_{\text{PR}_r}$ and $n_{\text{SR}_b}$ denote Gaussian noises at $\text{PR}_r$ and $\text{SR}_b$, respectively. For the ease of presentation and analysis, we can assume that the Gaussian noises at all receivers B have zero mean and unit variance. In addition, we transmit the power $P_{\text{SS}_1}$ and $P_{\text{SS}_2}$, which will be derived by closed-form expressions in the next section.

From (6), we can formulate the instantaneous signal-to-interference-plus-noise ratio (SINR) obtained at $\text{PR}_r$ as

$$\psi_{\text{PT}_t,\text{PR}_r} = \frac{P_{\text{PT}} g_{\text{PT}_t\text{PR}_r}}{P_{\text{SS}_1} g_{\text{SS}_1\text{PR}_r} + P_{\text{SS}_2} g_{\text{SS}_2\text{PR}_r} + 1}. \tag{8}$$

From (8), the TAS/SC algorithm can be written as follows:

$$(t, r) : \psi_{\text{PT}_t,\text{PR}_r} = \max_{p=1,2,\ldots,N_{\text{T}}} \max_{q=1,2,\ldots,N_{\text{R}}} \left( \psi_{\text{PT}_p,\text{PR}_q} \right). \tag{9}$$

Equation (9) implies that PT and PR cooperate to choose the best transmit and receive antennas to maximize the SINR obtained in this time slot.

For $\text{SR}_b$, since $d_{\text{SS}_1\text{SR}} \leq d_{\text{SS}_2\text{SR}}$, and hence $P_{\text{SS}_1} \geq P_{\text{SS}_2}$ (see Section 3.2), and $\text{SR}_b$ has to decode $x_1$ first. From (7), the SINR obtained for decoding $x_1$ is calculated as

$$\psi_{\text{SR}_b,x_1} = \frac{P_{\text{SS}_1} g_{\text{SS}_1\text{SR}_b}}{P_{\text{SS}_2} g_{\text{SS}_2\text{SR}_b} + P_{\text{PT}} g_{\text{PT}_t\text{SR}_b} + 1}. \tag{10}$$

Then, if $x_1$ can be correctly decoded, $\text{SR}_b$ can be removed $\sqrt{P_{\text{SS}_1}} h_{\text{SS}_1\text{SR}_b} x_1$ in $y_{\text{SR}_b}$ [20–25]. Then, the signal used to decode $x_2$ is given as

$$y_{\text{SR}_b}^* = \sqrt{P_{\text{SS}_2}} h_{\text{SS}_2\text{SR}_b} x_2 + \sqrt{P_{\text{PT}}} h_{\text{PT}_t\text{SR}_b} x_{\text{P}} + n_{\text{SR}_b}. \tag{11}$$

From (11), the SINR obtained for decoding $x_2$ is written as

$$\psi_{\text{SR}_b,x_2} = \frac{P_{\text{SS}_2} g_{\text{SS}_2\text{SR}_b}}{P_{\text{PT}} g_{\text{PT}_t\text{SR}_b} + 1}. \tag{12}$$

**Remark 3.** *If $SR_b$ only correctly decodes $x_1$, it will only send $x_1$ to $SS_2$ at the second time slot. It is worth noting that if $SR_b$ cannot successfully decode $x_1$, it cannot perform ICT to remove $x_1$, and hence $x_2$ is also not decoded. Moreover, in the case where $SR_b$ cannot correctly obtain $x_1$ and $x_2$, it will do nothing at the second time slot.*

Let us consider the secondary time slot, and assume that $\text{SR}_b$ can access the licensed band to transmit the data $x^*$ ($x^* \in \{x_\oplus, x_1\}$). In this time slot, assume that PT uses the $w$-th antenna to send the $x_{\text{P}}$ to PR. Then, the received signals at the $z$-th antenna of PR, at $\text{SS}_1$ and at $\text{SS}_2$ can be written, respectively, as

$$y_{\text{PR}_z} = \sqrt{P_{\text{PT}}} h_{\text{PT}_w\text{PR}_z} x_{\text{P}} + \sqrt{P_{\text{SR}_b}} h_{\text{SR}_b\text{PR}_z} x^* + n_{\text{PR}_z}, \tag{13}$$

$$y_{\text{SS}_1} = \sqrt{P_{\text{SR}_b}} h_{\text{SR}_b\text{SS}_1} x^* + \sqrt{P_{\text{PT}}} h_{\text{PT}_w\text{SS}_1} x_{\text{P}} + n_{\text{SS}_1}, \tag{14}$$

$$y_{\text{SS}_2} = \sqrt{P_{\text{SR}_b}} h_{\text{SR}_b\text{SS}_2} x^* + \sqrt{P_{\text{PT}}} h_{\text{PT}_w\text{SS}_2} x_{\text{P}} + n_{\text{SS}_2}, \tag{15}$$

where $P_{\text{SR}_b}$ is the transmit power of $\text{SR}_b$ ($P_{\text{SR}_b}$ which will be derived in the next section), and $n_{\text{B}}$ denotes the Gaussian noises at the receiver B whose zero mean and unit variance with $\text{B} \in \{\text{PR}_z, \text{SS}_1, \text{SS}_2\}$.

From (13), the SINR received at $PR_z$ is calculated as

$$\varphi_{PT_w,PR_z} = \frac{P_{PT}g_{PT_wPR_z}}{P_{SR_b}g_{SR_bPR_z} + 1}. \tag{16}$$

Similarly to (9), the TAS/SC algorithm in the second time slot can be written as

$$(w,z) : \varphi_{PT_w,PR_z} = \max_{p=1,2,\dots,N_T} \max_{q=1,2,\dots,N_R} \left( \varphi_{PT_p,PR_q} \right). \tag{17}$$

**Remark 4.** *To realize the TAS/SC algorithms in (9) and (17), in the set-up phase, PT, $SS_1$, $SS_2$, and $SR_b$ send pilot signals to PR. Then, PR estimates the channel coefficients $h_{PT_pPR_q}$, $h_{SS_1PR_q}$, $h_{SS_2PR_q}$, and $h_{SR_bPR_q}$ to calculate the instantaneous SINRs $\varphi_{PT_p,PR_q}$. Using (9) and (17), PR can determine the best transmitting antennas at PT and its best receiving antennas, in the first and secondary slots. Finally, PR feedbacks the index of these antennas to PT. Here, we also assume that the CSI estimation at PR is perfect.*

It is worth noting that various diversity transmitting/receiving techniques such as TAS/maximal ratio combining (MRC), maximal ratio transmission (MRT)/SC, and MRT/MRC can be used to enhance the OP performance for the primary network. However, the MRT and MRC techniques require both the amplitude and phase information of the channel coefficients. As a result, the implementation of the MRT and MRC techniques is more complex than that of TAS and SC. Moreover, as with using the MRT technique, PT has to use all its transmit antennas, which can cause more co-channel interference on the secondary network.

Next, for $SS_1$ and $SS_2$, the SINRs obtained for decoding $x^*$ can be formulated, respectively, as

$$\varphi_{SS_1} = \frac{P_{SR_b}g_{SR_bSS_1}}{P_{PT}g_{PT_wSS_1} + 1}, \varphi_{SS_2} = \frac{P_{SR_b}g_{SR_bSS_2}}{P_{PT}g_{PT_wSS_2} + 1}. \tag{18}$$

We note from (18) that when $x^* \equiv x_1$, this means that $SR_b$ only sends $x_1$ to $SS_2$ at the second time slot, and hence $SS_1$ is the outage in this case. When $x^* \equiv x_\oplus$, the secondary source $SS_i(i = 1, 2)$ attempts to decode $x_\oplus$ so that it can obtain the desired data by using the XOR rule, i.e., $x_i \oplus x_\oplus = x_j$, where $j = 1, 2$ and $j \neq i$.

This paper evaluates OP for the primary and secondary networks. For the primary network, OP in the first and second time slots can be formulated, respectively, as

$$OP_{P1} = Pr(\psi_{PT_t,PR_r} < \theta_{Pth}), OP_{P2} = Pr(\varphi_{PT_w,PR_z} < \theta_{Pth}), \tag{19}$$

where $\theta_{Pth}$ is a pre-determined threshold of the primary network.

For the secondary network, we consider OP at $SS_1$ and $SS_2$. At first, the OP of $SS_1$ is formulated as

$$OP_{SS_1} = 1 - Pr(\psi_{SR_b,x_1} \geq \theta_{Sth}, \psi_{SR_b,x_2} \geq \theta_{Sth}, \varphi_{SS_1} \geq \theta_{Sth}), \tag{20}$$

where $\theta_{Sth}$ is a pre-determined threshold of the secondary network.

In (20), $Pr(\psi_{SR_b,x_1} \geq \theta_{Sth}, \psi_{SR_b,x_2} \geq \theta_{Sth}, \varphi_{SS_1} \geq \theta_{Sth})$ is the probability that $SS_1$ can correctly receive the data $x_2$, i.e., $x_2$ is successfully obtained by $SR_b$ at the first time slot $(\psi_{SR_b,x_1} \geq \theta_{Sth}, \psi_{SR_b,x_2} \geq \theta_{Sth})$, and the transmission from $SR_b$ to $SS_1$ at the second time slot is successful $(\varphi_{SS_1} \geq \theta_{Sth})$.

Then, the OP of $SS_2$ can be formulated as

$$OP_{SS_2} = 1 - Pr(\psi_{SR_b,x_1} \geq \theta_{Sth}, \varphi_{SS_2} \geq \theta_{Sth}), \tag{21}$$

where $Pr(\psi_{SR_b,x_1} \geq \theta_{Sth}, \varphi_{SS_2} \geq \theta_{Sth})$ is the probability that $x_1$ is successfully decoded by $SR_b$ and $SS_2$ at the first and second time slots, respectively.

## 3. Performance Analysis

At first, we evaluate the OP of the primary network, and use this result to calculate the transmit power of $SS_1$, $SS_2$ and $SR_b$.

### 3.1. OP of the Primary Network

Combining (8), (9), and (19), the OP at the first time slot can be formulated as

$$
\begin{aligned}
OP_{P1} &= \Pr\left( \max_{p=1,2,\dots,N_T} \max_{q=1,2,\dots,N_R} \left( \psi_{PT_p,PR_q} \right) < \theta_{Pth} \right) \\
&= \left[ \Pr\left( \psi_{PT_p,PR_q} < \theta_{Pth} \right) \right]^{N_T N_R} \\
&= \left[ \underbrace{\Pr\left( g_{PT_p PR_q} < \rho_1 g_{SS_1 PR_q} + \rho_2 g_{SS_2 PR_q} + \rho_0 \right)}_{\mathcal{I}_1} \right]^{N_T N_R},
\end{aligned}
\tag{22}
$$

where

$$
\rho_0 = \frac{\theta_{Pth}}{P_{PT}}, \quad \rho_1 = \frac{P_{SS_1}\theta_{Pth}}{P_{PT}}, \quad \rho_2 = \frac{P_{SS_2}\theta_{Pth}}{P_{PT}}.
$$

As marked in (22), the probability $\mathcal{I}_1$ can be expressed in the following form:

$$
\mathcal{I}_1 = \int_0^{+\infty} \int_0^{+\infty} F_{g_{PT_p PR_q}}(\rho_1 x + \rho_2 y + \rho_0) f_{g_{SS_1 PR_q}}(x) f_{g_{SS_2 PR_q}}(y) \, dx \, dy.
\tag{23}
$$

Substituting CDF $F_{g_{PT_p PR_q}}(\rho_1 x + \rho_2 y + \rho_0)$ and PDFs $f_{g_{SS_1 PR_q}}(x)$ and $f_{g_{SS_2 PR_q}}(y)$ into (23); after some careful calculation, we obtain an exact closed-form expression of $\mathcal{I}_1$. Then, substituting this result into (22) yields

$$
OP_{P1} = \left[ 1 - \frac{\Omega_{SS_1 PR}}{\Omega_{SS_1 PR} + \Omega_{PTPR}\rho_1} \frac{\Omega_{SS_2 PR}}{\Omega_{SS_2 PR} + \Omega_{PTPR}\rho_2} \exp(-\Omega_{PTPR}\rho_0) \right]^{N_T N_R}.
\tag{24}
$$

Similarly, combining (16), (17), and (19), the OP of the primary network at the second time slot can be exactly computed as

$$
\begin{aligned}
OP_{P2} &= \Pr\left( \max_{p=1,2,\dots,N_T} \max_{q=1,2,\dots,N_R} \left( \varphi_{PT_p,PR_q} \right) < \theta_{Pth} \right) \\
&= \left[ \Pr\left( \varphi_{PT_p,PR_q} < \theta_{Pth} \right) \right]^{N_T N_R} \\
&= \left[ \Pr\left( g_{PT_p PR_q} < \rho_3 g_{SR_b PR_q} + \rho_0 \right) \right]^{N_T N_R} \\
&= \left[ 1 - \frac{\Omega_{SRPR}}{\Omega_{SRPR} + \Omega_{PTPR}\rho_3} \exp(-\Omega_{PTPR}\rho_0) \right]^{N_T N_R},
\end{aligned}
\tag{25}
$$

where

$$
\rho_3 = \frac{P_{SR_b}\theta_{Pth}}{P_{PT}}.
$$

**Remark 5**. *We now consider a special case where the secondary users are not currently using the licensed band. Because there is no co-channel interference from the secondary network, the TAS/SC algorithms in (9) and (17) can be re-written, under the following form:*

$$
(k,l): g_{PT_k PR_l} = \max_{p=1,2,\dots,N_T} \max_{q=1,2,\dots,N_R} \left( g_{PT_p PR_q} \right).
\tag{26}
$$

From (26), it is straightforward to calculate the OP of the primary network as

$$\text{OP}_{\text{P0}} = \Pr\big(P_{\text{PT}}g_{\text{PT}_k\text{PR}_l} < \theta_{\text{Pth}}\big)$$

$$= \Pr\left(\max_{p=1,2,\dots,N_\text{T}} \max_{q=1,2,\dots,N_\text{R}} \big(g_{\text{PT}_p\text{PR}_q}\big) < \rho_0\right)$$

$$= \left[1 - \exp(-\Omega_{\text{PTPR}}\rho_0)\right]^{N_\text{T}N_\text{R}}. \tag{27}$$

*3.2. Transmit Power of* $\text{SS}_1$, $\text{SS}_2$ *and* $\text{SR}_b$

As proposed in [7–9], let $\varepsilon_{\text{OP}}$ denote the target QoS of the primary network, i.e., $\text{OP}_{\text{P1}} \leq \varepsilon_{\text{OP}}$ and $\text{OP}_{\text{P2}} \leq \varepsilon_{\text{OP}}$. To find the transmit power of $\text{SS}_1$, $\text{SS}_2$, and $\text{SR}_b$, we have to solve the following equations: $\text{OP}_{\text{P1}} = \varepsilon_{\text{OP}}$ and $\text{OP}_{\text{P2}} = \varepsilon_{\text{OP}}$. Then, using (25) to solve $\text{OP}_{\text{P2}} = \varepsilon_{\text{OP}}$, we have

$$\rho_3 = (\Delta - 1)\frac{\Omega_{\text{SRPR}}}{\Omega_{\text{PTPR}}} \Rightarrow P_{\text{SR}_b} = \frac{\Delta - 1}{\theta_{\text{Pth}}}\frac{\Omega_{\text{SRPR}}}{\Omega_{\text{PTPR}}}P_{\text{PT}}, \tag{28}$$

where

$$\Delta = \frac{1}{1 - (\varepsilon_{\text{OP}})^{\frac{1}{N_\text{T}N_\text{R}}}} \exp(-\Omega_{\text{PTPR}}\rho_0). \tag{29}$$

Because $P_{\text{SR}_b}$ is not negative, Equation (28) is re-written as follows:

$$P_{\text{SR}_b} = \left[\frac{\Delta - 1}{\theta_{\text{Pth}}}\frac{\Omega_{\text{SRPR}}}{\Omega_{\text{PTPR}}}P_{\text{PT}}\right]^+, \tag{30}$$

where $[x]^+ = \max(0, x)$.

From (27)–(30), it is straightforward that $P_{\text{SR}_b} > 0$ if

$$\Delta - 1 > 0 \Leftrightarrow \left[1 - \exp(-\Omega_{\text{PTPR}}\rho_0)\right]^{N_\text{T}N_\text{R}} < \varepsilon_{\text{OP}}$$

$$\Leftrightarrow \text{OP}_{\text{P0}} < \varepsilon_{\text{OP}}. \tag{31}$$

**Remark 6.** *Equation (31) implies that the primary network only shares the licensed band with the secondary network if the OP of the primary network without the interference from the secondary network $(OP_{P0})$ is less than the required QoS. Otherwise, if $OP_{P0} \geq \varepsilon_{OP}$, then $SR_b$ is not allowed to access the licensed band, i.e., $P_{SR_b} = 0$. It is also noted from (30) that the transmit power of the secondary relays is the same.*

Then, we consider $P_{\text{SS}_1}$ and $P_{\text{SS}_2}$. Similarly to the power allocation method proposed in [7,8], we have the following formula:

$$\frac{P_{\text{SS}_1}}{\Omega_{\text{SS}_1\text{PR}}} = \frac{P_{\text{SS}_2}}{\Omega_{\text{SS}_2\text{PR}}} = \delta. \tag{32}$$

Equation (32) presents that the average interference power at PR (due to the data transmission of $\text{SS}_1$ and $\text{SS}_2$) is the same or $P_{\text{SS}_1}\big(d_{\text{SS}_1\text{PR}}\big)^{-\mu} = P_{\text{SS}_2}\big(d_{\text{SS}_2\text{PR}}\big)^{-\mu}$. Therefore, if $d_{\text{SS}_1\text{PR}} \geq d_{\text{SS}_2\text{PR}}$, then $P_{\text{SS}_1} \geq P_{\text{SS}_2}$ and vice versa. Now, combining (24) and (32), we obtain

$$\text{OP}_{\text{P1}} = \left[1 - \left(\frac{P_{\text{PT}}}{P_{\text{PT}} + \Omega_{\text{PTPR}}\theta_{\text{Pth}}\delta}\right)^2 \exp(-\Omega_{\text{PTPR}}\rho_0)\right]^{N_\text{T}N_\text{R}}. \tag{33}$$

Then, solving $\text{OP}_{\text{P1}} = \varepsilon_{\text{OP}}$, we can find $P_{\text{SS}_1}$ and $P_{\text{SS}_2}$, respectively, as

$$P_{\text{SS}_1} = \left[ \frac{\sqrt{\Delta} - 1}{\theta_{\text{Pth}}} \frac{\Omega_{\text{SS}_1\text{PR}}}{\Omega_{\text{PTPR}}} P_{\text{PT}} \right]^+, P_{\text{SS}_2} = \left[ \frac{\sqrt{\Delta} - 1}{\theta_{\text{Pth}}} \frac{\Omega_{\text{SS}_2\text{PR}}}{\Omega_{\text{PTPR}}} P_{\text{PT}} \right]^+. \tag{34}$$

**Remark 7.** *Similarly, the condition for $P_{\text{SS}_1}, P_{\text{SS}_2} > 0$ is $\text{OP}_{\text{P0}} < \varepsilon_{\text{OP}}$. Moreover, as $P_{\text{PT}} \to +\infty$, we have $\exp(-\Omega_{\text{PTPR}}\rho_0) \approx 1$, and then we can approximate $P_{\text{SR}_b}, P_{\text{SS}_1}$, and $P_{\text{SS}_2}$, respectively, as*

$$P_{\text{SR}_b} \stackrel{P_{\text{PT}} \to +\infty}{\approx} \frac{\Delta^* - 1}{\theta_{\text{Pth}}} \frac{\Omega_{\text{SRPR}}}{\Omega_{\text{PTPR}}} P_{\text{PT}}, P_{\text{SS}_1} \stackrel{P_{\text{PT}} \to +\infty}{\approx} \frac{\left( \sqrt{\Delta^*} - 1 \right)}{\theta_{\text{Pth}}} \frac{\Omega_{\text{SS}_1\text{PR}}}{\Omega_{\text{PTPR}}} P_{\text{PT}},$$

$$P_{\text{SS}_2} \stackrel{P_{\text{PT}} \to +\infty}{\approx} \frac{\left( \sqrt{\Delta^*} - 1 \right)}{\theta_{\text{Pth}}} \frac{\Omega_{\text{SS}_2\text{PR}}}{\Omega_{\text{PTPR}}} P_{\text{PT}}. \tag{35}$$

*where $\Delta^* = \left[ 1 - (\varepsilon_{\text{OP}})^{\frac{1}{N_\text{T} N_\text{R}}} \right]^{-1}$. We can observe from (35) that at high transmit power $P_{\text{PT}}$, the transmit power $P_{\text{SR}_b}, P_{\text{SS}_1}$, and $P_{\text{SS}_2}$ linearly increases as $P_{\text{PT}}$ increases.*

### *3.3. OP of the Secondary Network*

This sub-section calculates $\text{OP}_{\text{SS}_2}$ and $\text{OP}_{\text{SS}_1}$ as in Propositions 1 and 2 below. At first, plugging (10), (12), and (18)–(21) together, we can rewrite $\text{OP}_{\text{SS}_2}$ and $\text{OP}_{\text{SS}_1}$, respectively, as

$$\text{OP}_{\text{SS}_2} = 1 - \underbrace{\Pr(X_1 \geq \alpha_1 X_2 + \alpha_2 U + \alpha_0, X_2 \geq \alpha_6 V_2 + \alpha_5)}_{\mathcal{I}_2}, \tag{36}$$

$$\text{OP}_{\text{SS}_1} = 1 - \underbrace{\Pr(X_1 \geq \alpha_1 X_2 + \alpha_2 U + \alpha_0, X_2 \geq \alpha_4 U + \alpha_3, X_1 \geq \alpha_6 V_1 + \alpha_5)}_{\mathcal{I}_3}, \tag{37}$$

where $X_1 = g_{\text{SS}_1\text{SR}_b} = g_{\text{SR}_b\text{SS}_1}$, $X_2 = g_{\text{SS}_2\text{SR}_b} = g_{\text{SR}_b\text{SS}_2}$, $U = g_{\text{PT}_t\text{SR}_b}$, $V_1 = g_{\text{PT}_w\text{SS}_1}$, $V_2 = g_{\text{PT}_w\text{SS}_2}$, and

$$\alpha_0 = \frac{\theta_{\text{Sth}}}{P_{\text{SS}_1}}, \alpha_1 = \frac{\theta_{\text{Sth}}P_{\text{SS}_2}}{P_{\text{SS}_1}}, \alpha_2 = \frac{\theta_{\text{Sth}}P_{\text{PT}}}{P_{\text{SS}_1}}, \alpha_3 = \frac{\theta_{\text{Sth}}}{P_{\text{SS}_2}}, \alpha_4 = \frac{P_{\text{PT}}\theta_{\text{Sth}}}{P_{\text{SS}_2}}, \alpha_5 = \frac{\theta_{\text{Sth}}}{P_{\text{SR}_b}}, \alpha_6 = \frac{P_{\text{PT}}\theta_{\text{Sth}}}{P_{\text{SR}_b}}.$$

**Proposition 1.** *$\text{OP}_{\text{SS}_2}$ can be expressed by an exact closed-form expression as*

$$\text{OP}_{\text{SS}_2} = 1 - \frac{\Omega_{\text{PTSR}}}{\Omega_{\text{PTSR}} + \Omega_{\text{SS}_1\text{SR}}\alpha_2} \exp\left(-\Omega_{\text{SS}_1\text{SR}}\alpha_0\right)$$

$$\times \sum_{p=0}^{M-1} (-1)^p \frac{C_{M-1}^p M \Omega_{\text{SS}_2\text{SR}}}{(p+1)\Omega_{\text{SS}_2\text{SR}} + \Omega_{\text{SS}_1\text{SR}}\alpha_1} \frac{\Omega_{\text{PTSS}_2}}{\Omega_{\text{PTSS}_2} + ((p+1)\Omega_{\text{SS}_2\text{SR}} + \Omega_{\text{SS}_1\text{SR}}\alpha_1)\alpha_6}$$

$$\times \exp\left(-((p+1)\Omega_{\text{SS}_2\text{SR}} + \Omega_{\text{SS}_1\text{SR}}\alpha_1)\alpha_5\right). \tag{38}$$

**Proof.** Setting $U = u$, $V_2 = v_2$, we can formulate $\mathcal{I}_2$ in (36), conditioned on $U = u$, $V_2 = v_2$, as

$$\mathcal{I}_2(u, v_2) = \int_{\alpha_6 v_2 + \alpha_5}^{+\infty} \left[ 1 - F_{X_1}(\alpha_1 x_2 + \alpha_2 u + \alpha_0) \right] f_{X_2}(x_2) dx_2. \tag{39}$$

Substituting the CDF of $X_1$ in (1) and PDF of $X_2$ in (5) into (39), after some calculation, we obtain

$$
\begin{aligned}
\mathcal{I}_2(u, v_2) ={}& \exp\left(-\Omega_{\mathrm{SS_1SR}}\alpha_0\right) \exp\left(-\Omega_{\mathrm{SS_1SR}}\alpha_2 u\right) \\
&\times \sum_{p=0}^{M-1} (-1)^p \frac{C_{M-1}^p M\Omega_{\mathrm{SS_2SR}}}{(p+1)\Omega_{\mathrm{SS_2SR}} + \Omega_{\mathrm{SS_1SR}}\alpha_1} \exp\left(-\left((p+1)\Omega_{\mathrm{SS_2SR}} + \Omega_{\mathrm{SS_1SR}}\alpha_1\right)\alpha_5\right) \\
&\times \exp\left(-\left((p+1)\Omega_{\mathrm{SS_2SR}} + \Omega_{\mathrm{SS_1SR}}\alpha_1\right)\alpha_6 v_2\right).
\end{aligned}
\tag{40}
$$

Then, from (40), we obtain an exact closed-form expression of $\mathcal{I}_2$ as follows:

$$
\begin{aligned}
\mathcal{I}_2 ={}& \int_0^{+\infty} \int_0^{+\infty} \mathcal{I}_2(u, v_2) f_U(u) f_{V_2}(v_2)\, du\, dv \\
={}& \frac{\Omega_{\mathrm{PTSR}}}{\Omega_{\mathrm{PTSR}} + \Omega_{\mathrm{SS_1SR}}\alpha_2} \exp\left(-\Omega_{\mathrm{SS_1SR}}\alpha_0\right) \\
&\times \sum_{p=0}^{M-1} (-1)^p \frac{C_{M-1}^p M\Omega_{\mathrm{SS_2SR}}}{(p+1)\Omega_{\mathrm{SS_2SR}} + \Omega_{\mathrm{SS_1SR}}\alpha_1} \frac{\Omega_{\mathrm{PTSS_2}}}{\Omega_{\mathrm{PTSS_2}} + \left((p+1)\Omega_{\mathrm{SS_2SR}} + \Omega_{\mathrm{SS_1SR}}\alpha_1\right)\alpha_6} \\
&\times \exp\left(-\left((p+1)\Omega_{\mathrm{SS_2SR}} + \Omega_{\mathrm{SS_1SR}}\alpha_1\right)\alpha_5\right).
\end{aligned}
\tag{41}
$$

Substituting (41) into (36), we obtain $\mathrm{OP}_{\mathrm{SS_2}}$, and finish the proof. $\square$

**Proposition 2.** $\mathrm{OP}_{\mathrm{SS_1}}$ *can be expressed as follows:*

- **Case 1:** $P_{\mathrm{SS_1}} > (1 + \theta_{\mathrm{Sth}}) P_{\mathrm{SR}_b}$

$$
\begin{aligned}
\mathrm{OP}_{\mathrm{SS_1}} ={}& 1 - \sum_{p=1}^M \left( \frac{B_1 \Omega_{\mathrm{PTSS_1}}}{\Omega_{\mathrm{PTSS_1}} + \beta_1} - \frac{B_2 \Omega_{\mathrm{PTSS_1}}}{\Omega_{\mathrm{PTSS_1}} + \beta_3} - \frac{B_3 \Omega_{\mathrm{PTSS_1}}}{\Omega_{\mathrm{PTSS_1}} + \beta_2} + \frac{B_4 \Omega_{\mathrm{PTSS_1}}}{\Omega_{\mathrm{PTSS_1}} + \beta_4} \right) \\
&- \sum_{p=1}^M A_3 \frac{\Omega_{\mathrm{PTSR}}}{\Omega_{\mathrm{PTSR}} + \chi_3} \cdot \frac{\Omega_{\mathrm{PTSS_1}}\alpha_8}{\Omega_{\mathrm{PTSS_1}}\alpha_8 + (\Omega_{\mathrm{PTSR}} + \chi_3)\alpha_6} \exp\left(-(\Omega_{\mathrm{PTSR}} + \chi_3)\frac{\alpha_5 - \alpha_7}{\alpha_8}\right) \\
&+ \sum_{p=1}^M A_4 \frac{\Omega_{\mathrm{PTSR}}}{\Omega_{\mathrm{PTSR}} + \chi_4} \frac{\Omega_{\mathrm{PTSS_1}}\alpha_8}{\Omega_{\mathrm{PTSS_1}}\alpha_8 + (\Omega_{\mathrm{PTSR}} + \chi_4)\alpha_6} \exp\left(-(\Omega_{\mathrm{PTSR}} + \chi_4)\frac{\alpha_5 - \alpha_7}{\alpha_8}\right),
\end{aligned}
\tag{42}
$$

- **Case 2:** $P_{\mathrm{SS_1}} \le (1 + \theta_{\mathrm{Sth}}) P_{\mathrm{SR}_b}$

$$
\begin{aligned}
\mathrm{OP}_{\mathrm{SS_1}} ={}& 1 - \sum_{p=1}^M \left[ \frac{C_1 \Omega_{\mathrm{PTSR}}}{\Omega_{\mathrm{PTSR}} + \beta_5} - \frac{C_2 \Omega_{\mathrm{PTSR}}}{\Omega_{\mathrm{PTSR}} + \beta_6} \right] \\
&- \sum_{p=1}^M \frac{A_3 \Omega_{\mathrm{PTSR}}}{\Omega_{\mathrm{PTSR}} + \chi_3} - \sum_{p=1}^M \frac{A_3 \Omega_{\mathrm{PTSR}}\alpha_6}{(\Omega_{\mathrm{PTSR}} + \chi_3)\alpha_6 + \Omega_{\mathrm{PTSS_1}}\alpha_8} \exp\left(-\Omega_{\mathrm{PTSS_1}}\frac{\alpha_7 - \alpha_5}{\alpha_6}\right) \\
&- \sum_{p=1}^M \frac{A_4 \Omega_{\mathrm{PTSR}}}{\Omega_{\mathrm{PTSR}} + \chi_4} + \sum_{p=1}^M \frac{A_4 \Omega_{\mathrm{PTSR}}\alpha_6}{(\Omega_{\mathrm{PTSR}} + \chi_4)\alpha_6 + \Omega_{\mathrm{PTSS_1}}\alpha_8} \exp\left(-\Omega_{\mathrm{PTSS_1}}\frac{\alpha_7 - \alpha_5}{\alpha_6}\right),
\end{aligned}
\tag{43}
$$

where

$$A_2 = (-1)^{p+1} C_M^p \frac{\Omega_{SS_1SR}\alpha_1}{\Omega_{SS_1SR}\alpha_1 + p\Omega_{SS_2SR}} \exp\left(\frac{p\Omega_{SS_2SR}\alpha_0 - (\Omega_{SS_1SR}\alpha_1 + p\Omega_{SS_2SR})\alpha_6}{\alpha_1}\right),$$

$$\beta_1 = \Omega_{SS_1SR}\alpha_6, \beta_2 = \frac{(\Omega_{SS_1SR}\alpha_1 + p\Omega_{SS_2SR})\alpha_6}{\alpha_1}, \chi_1 = p\Omega_{SS_2SR}\alpha_4, \chi_2 = \frac{p\Omega_{SS_2SR}\alpha_2}{\alpha_1},$$

$$A_3 = (-1)^{p+1} C_M^p \exp\left(-\Omega_{SS_1SR}\alpha_7 - p\Omega_{SS_2SR}\alpha_3\right),$$

$$A_4 = (-1)^{p+1} C_M^p \frac{\Omega_{SS_1SR}\alpha_1}{\Omega_{SS_1SR}\alpha_1 + p\Omega_{SS_2SR}} \exp\left(\frac{p\Omega_{SS_2SR}\alpha_0 - (\Omega_{SS_1SR}\alpha_1 + p\Omega_{SS_2SR})\alpha_7}{\alpha_1}\right),$$

$$\chi_3 = \Omega_{SS_1SR}\alpha_8 + p\Omega_{SS_2SR}\alpha_4, \chi_4 = \frac{(\Omega_{SS_1SR}\alpha_1 + p\Omega_{SS_2SR})\alpha_8 - p\Omega_{SS_2SR}\alpha_2}{\alpha_1},$$

$$B_1 = \frac{A_1\Omega_{PTSR}}{\Omega_{PTSR} + \chi_1}, B_2 = \frac{A_1\Omega_{PTSR}}{\Omega_{PTSR} + \chi_1} \exp\left(-\frac{(\Omega_{PTSR} + \chi_1)(\alpha_5 - \alpha_7)}{\alpha_8}\right)$$

$$B_3 = \frac{A_2\Omega_{PTSR}}{\Omega_{PTSR} + \chi_2}, B_4 = \frac{A_2\Omega_{PTSR}}{\Omega_{PTSR} + \chi_2} \exp\left(-\frac{(\Omega_{PTSR} + \chi_2)(\alpha_5 - \alpha_7)}{\alpha_8}\right)$$

$$\beta_3 = \frac{(\Omega_{PTSR} + \chi_1)\alpha_6}{\alpha_8}, \beta_4 = \frac{(\Omega_{PTSR} + \chi_2)\alpha_6}{\alpha_8},$$

$$C_1 = \frac{A_1\Omega_{PTSS_1}}{\Omega_{PTSS_1} + \beta_1} \exp\left(-\frac{(\Omega_{PTSS_1} + \beta_1)(\alpha_7 - \alpha_5)}{\alpha_6}\right), \beta_5 = \frac{\chi_1\alpha_6 + (\Omega_{PTSS_1} + \beta_1)\alpha_8}{\alpha_6}$$

$$C_2 = \frac{A_2\Omega_{PTSS_1}}{\Omega_{PTSS_1} + \beta_2} \exp\left(-\frac{(\Omega_{PTSS_1} + \beta_2)(\alpha_7 - \alpha_5)}{\alpha_6}\right), \beta_6 = \frac{-\chi_2\alpha_6 + (\Omega_{PTSS_1} + \beta_2)\alpha_8}{\alpha_6}.$$

**Proof.** At first, by setting $U = u$, $V_1 = v_1$, we can write $\mathcal{I}_3$ in (37), conditioned on $U = u$, $V_2 = v_2$, as

$$\mathcal{I}_3(u, v_1) = \Pr(X_1 \geq \alpha_1 X_2 + \alpha_2 u + \alpha_0, X_2 \geq \alpha_4 u + \alpha_3, X_1 \geq \alpha_6 v_1 + \alpha_5). \quad (44)$$

Then, $\mathcal{I}_3(u, v_1)$ can be formulated in two conditions: (C1) $\alpha_8 u + \alpha_7 < \alpha_6 v_1 + \alpha_5$; and (C2) $\alpha_8 u + \alpha_7 \geq \alpha_6 v_1 + \alpha_5$, respectively, as

$$\mathcal{I}_3(u, v_1|C_1) = \int_{\alpha_6 v_1 + \alpha_5}^{+\infty} f_{X_1}(x_1) \left[\int_{\alpha_4 u + \alpha_3}^{\frac{x_1 - \alpha_2 u - \alpha_0}{\alpha_1}} f_{X_2}(x_2) dx_2\right] dx_1$$

$$= \int_{\alpha_6 v_1 + \alpha_5}^{+\infty} f_{X_1}(x_1) \left[F_{X_2}\left(\frac{x_1 - \alpha_2 u - \alpha_0}{\alpha_1}\right) - F_{X_2}(\alpha_4 u + \alpha_3)\right] dx_1, \quad (45)$$

$$\mathcal{I}_3(u, v_1|C_2) = \int_{\alpha_8 u + \alpha_7}^{+\infty} f_{X_1}(x_1) \left[\int_{\alpha_4 u + \alpha_3}^{\frac{x_1 - \alpha_2 u - \alpha_0}{\alpha_1}} f_{X_2}(x_2) dx_2\right] dx_1$$

$$= \int_{\alpha_8 u + \alpha_7}^{+\infty} f_{X_1}(x_1) \left[F_{X_2}\left(\frac{x_1 - \alpha_2 u - \alpha_0}{\alpha_1}\right) - F_{X_2}(\alpha_4 u + \alpha_3)\right] dx_1. \quad (46)$$

Substituting the PDF of $X_1$ given in (1), and CDF of $X_2$ given in (4) into (45), after some calculation, we obtain a closed-form expression of $\mathcal{I}_3(u, v_1|C_1)$ as follows:

$$\mathcal{I}_3(u, v_1|C_1) = \sum_{p=1}^{M} A_1 \exp(-\beta_1 v_1) \exp(-\chi_1 u) - \sum_{p=1}^{M} A_2 \exp(-\beta_2 v_1) \exp(\chi_2 u). \quad (47)$$

For $\mathcal{I}_3(u, v_1|C_2)$ in (46), similarly, we have

$$\mathcal{I}_3(u, v_1|C_2) = \sum_{p=1}^{M} A_3 \exp(-\chi_3 u) - \sum_{p=1}^{M} A_4 \exp(-\chi_4 u). \tag{48}$$

Then, by averaging $\mathcal{I}_3(u, v_1)$, with respect to $U$ and $V_1$, we can obtain $\mathcal{I}_3$ in Case 1 and Case 2 as

- **Case 1**: $P_{\mathrm{SS}_1} > (1 + \theta_{\mathrm{Sth}})P_{\mathrm{SR}_b}$ or $\alpha_5 > \alpha_7$

  In this case, we have

$$\mathcal{I}_3 = \underbrace{\int_0^{+\infty} f_{V_1}(v_1) \left[ \int_0^{\frac{\alpha_6 v_1 + \alpha_5 - \alpha_7}{\alpha_8}} f_U(u) \mathcal{I}_3(u, v_1|C_1) du \right] dv_1}_{\mathcal{J}_1}$$

$$+ \underbrace{\int_0^{+\infty} f_{V_1}(v_1) \left[ \int_{\frac{\alpha_6 v_1 + \alpha_5 - \alpha_7}{\alpha_8}}^{+\infty} f_U(u) \mathcal{I}_3(u, v_1|C_2) du \right] dv_1}_{\mathcal{J}_2}. \tag{49}$$

Considering $\mathcal{J}_1$ as marked in (49); combining (1) and (45), we can write

$$\mathcal{J}_1 = \int_0^{+\infty} f_{V_1}(v_1) \left[ \begin{array}{c} \sum\limits_{p=1}^{M} B_1 \exp(-\beta_1 v_1) - \sum\limits_{p=1}^{M} B_2 \exp(-\beta_3 v_1) - \\ - \sum\limits_{p=1}^{M} B_3 \exp(-\beta_2 v_1) + \sum\limits_{p=1}^{M} B_4 \exp(-\beta_4 v_1) \end{array} \right] dv_1. \tag{50}$$

Next, substituting the PDF of $V_1$ given in (1) into (50), we obtain $\mathcal{J}_1$ as

$$\mathcal{J}_1 = \sum_{p=1}^{M} \left( \frac{B_1 \Omega_{\mathrm{PTSS}_1}}{\Omega_{\mathrm{PTSS}_1} + \beta_1} - \frac{B_2 \Omega_{\mathrm{PTSS}_1}}{\Omega_{\mathrm{PTSS}_1} + \beta_3} - \frac{B_3 \Omega_{\mathrm{PTSS}_1}}{\Omega_{\mathrm{PTSS}_1} + \beta_2} + \frac{B_4 \Omega_{\mathrm{PTSS}_1}}{\Omega_{\mathrm{PTSS}_1} + \beta_4} \right). \tag{51}$$

Similarly, we can exactly calculate $\mathcal{J}_2$ in (49) as

$$\mathcal{J}_2 = \sum_{p=1}^{M} A_3 \frac{\Omega_{\mathrm{PTSR}}}{\Omega_{\mathrm{PTSR}} + \chi_3} \cdot \frac{\Omega_{\mathrm{PTSS}_1} \alpha_8}{\Omega_{\mathrm{PTSS}_1} \alpha_8 + (\Omega_{\mathrm{PTSR}} + \chi_3)\alpha_6} \exp\left( -(\Omega_{\mathrm{PTSR}} + \chi_3)\frac{\alpha_5 - \alpha_7}{\alpha_8} \right)$$

$$- \sum_{p=1}^{M} A_4 \frac{\Omega_{\mathrm{PTSR}}}{\Omega_{\mathrm{PTSR}} + \chi_4} \frac{\Omega_{\mathrm{PTSS}_1} \alpha_8}{\Omega_{\mathrm{PTSS}_1} \alpha_8 + (\Omega_{\mathrm{PTSR}} + \chi_4)\alpha_6} \exp\left( -(\Omega_{\mathrm{PTSR}} + \chi_4)\frac{\alpha_5 - \alpha_7}{\alpha_8} \right). \tag{52}$$

- **Case 2**: $P_{\mathrm{SS}_1} \leq (1 + \theta_{\mathrm{Sth}})P_{\mathrm{SR}_b}$ or $\alpha_5 \leq \alpha_7$

  Similarly to Case 1, we can formulate $\mathcal{I}_3$ in this case as follows:

$$\mathcal{I}_3 = \underbrace{\int_0^{+\infty} f_U(u) \left[ \int_{\frac{\alpha_8 u + \alpha_7 - \alpha_5}{\alpha_6}}^{+\infty} f_{V_1}(v_1) \mathcal{I}_3(u, v_1|C_1) dv_1 \right] du}_{\mathcal{J}_3}$$

$$+ \underbrace{\int_0^{+\infty} f_U(u) \left[ \int_0^{\frac{\alpha_8 u + \alpha_7 - \alpha_5}{\alpha_6}} f_{V_1}(v_1) \mathcal{I}_3(u, v_1|C_2) dv_1 \right] du}_{\mathcal{J}_4}. \tag{53}$$

Similarly to the derivation of $\mathcal{J}_1$ and $\mathcal{J}_2$, we can obtain $\mathcal{J}_3$ and $\mathcal{J}_4$ in (53), respectively, as

$$\mathcal{J}_3 = \sum_{p=1}^{M} \left[ \frac{C_1 \Omega_{\text{PTSR}}}{\Omega_{\text{PTSR}} + \beta_5} - \frac{C_2 \Omega_{\text{PTSR}}}{\Omega_{\text{PTSR}} + \beta_6} \right], \tag{54}$$

$$\mathcal{J}_4 = \sum_{p=1}^{M} \frac{A_3 \Omega_{\text{PTSR}}}{\Omega_{\text{PTSR}} + \chi_3} - \sum_{p=1}^{M} \frac{A_3 \Omega_{\text{PTSR}} \alpha_6}{(\Omega_{\text{PTSR}} + \chi_3)\alpha_6 + \Omega_{\text{PTSS}_1}\alpha_8} \exp\left( -\Omega_{\text{PTSS}_1} \frac{\alpha_7 - \alpha_5}{\alpha_6} \right)$$

$$- \sum_{p=1}^{M} \frac{A_4 \Omega_{\text{PTSR}}}{\Omega_{\text{PTSR}} + \chi_4} + \sum_{p=1}^{M} \frac{A_4 \Omega_{\text{PTSR}} \alpha_6}{(\Omega_{\text{PTSR}} + \chi_4)\alpha_6 + \Omega_{\text{PTSS}_1}\alpha_8} \exp\left( -\Omega_{\text{PTSS}_1} \frac{\alpha_7 - \alpha_5}{\alpha_6} \right). \tag{55}$$

Substituting (49), (51), and (52) into (37), and substituting (53)–(55) into (37), we obtain $\text{OP}_{\text{SS}_1}$ in Case 1 and in Case 2, respectively. Therefore, we finish the proof here. □

## 4. Simulation Results

In a simulation environment, we the fix positions of $\text{SS}_1$, $\text{SS}_2$, PT, and PR at (0,0), (1,0), (0.65,1), and (0.65,0.5), respectively, while that of the secondary relays is $(x_\text{R}, 0)$, and where $0 < x_\text{R} \leq 0.5$. With these positions, we can see that $d_{\text{SS}_1\text{SR}} \leq d_{\text{SS}_2\text{SR}}$ and $d_{\text{SS}_1\text{PR}} > d_{\text{SS}_2\text{PR}}$. In this section, the system parameters are fixed as follows: the path-loss exponent equals 3 $(\mu = 3)$, the outage threshold of the primary network equals 1 $(\theta_{\text{Pth}} = 1)$, the target QoS of the primary network equals 0.001 $(\varepsilon_{\text{OP}} = 0.001)$, and the outage threshold of the secondary network equals 0.001 $(\theta_{\text{Sth}} = 0.001)$. Finally, in the figures presented below, the markers denote the simulated results (Sim), and the solid lines denote the theoretical results (Theory).

### 4.1. OP of the Primary Network and Transmit Power of the Secondary Transmitters

Figures 2 and 3, respectively, present the OP of the primary network and transmit power of the secondary transmitters as a function of $P_{\text{PT}}$ in dB. In these figures, the secondary nodes are located at (0.3,0), i.e., $x_\text{R} = 0.3$. At first, we can see from these figures that at low $P_{\text{PT}}$ values, the QoS of the primary network is not satisfied (i.e., $\text{OP}_{\text{P0}} > \varepsilon_{\text{OP}}$), and hence the secondary transmitters are not allowed to access the spectrum, (i.e., $P_{\text{SS}_1} = P_{\text{SS}_2} = P_{\text{SR}_b} = 0$). It is worth noting that without the secondary operation, the OP values at PR equals to $\text{OP}_{\text{P0}}$. Then, let us consider the case where the transmit power $P_{\text{PT}}$ is high enough and the QoS of the primary network is satisfied. In this case, $\text{SR}_b$, $\text{SS}_1$ and $\text{SS}_2$ can access the licensed band to transmit the data, using the maximum transmit power as given in (30) and (34). Therefore, the OP at PR is equal to $\varepsilon_{\text{OP}}$, i.e., $\text{OP}_{\text{P1}} = \text{OP}_{\text{P2}} = \varepsilon_{\text{OP}}$. As seen, when $N_\text{T} = 1$ and $N_\text{R} = 3$, the secondary network can access the licensed band when $P_{\text{PT}} \geq 1 (\text{dB})$, and when $N_\text{T} = N_\text{R} = 2$, the secondary network can access the licensed band when $P_{\text{PT}} \geq -1 (\text{dB})$. We also see that $N_\text{T} = N_\text{R} = 2$ provides better OP performance for the primary network as well as increases the spectrum access possibility for the secondary network, as compared with $N_\text{T} = 1$ and $N_\text{R} = 3$. Moreover, with $N_\text{T} = N_\text{R} = 2$, the transmit power of the secondary transmitters is also higher than those with $N_\text{T} = 1$ and $N_\text{R} = 3$. Looking at Figure 3, we also see that the transmit power of all secondary transmitters increases as $P_{\text{PT}}$ increases. Moreover, as proven in (35), the transmit power of all secondary transmitters linearly increases at high $P_{\text{PT}}$ region. From Figure 2, we can observe that the simulation results validate the theoretical ones of $\text{OP}_{\text{P1}}$, $\text{OP}_{\text{P2}}$, and $\text{OP}_{\text{P0}}$ derived in Section 3.

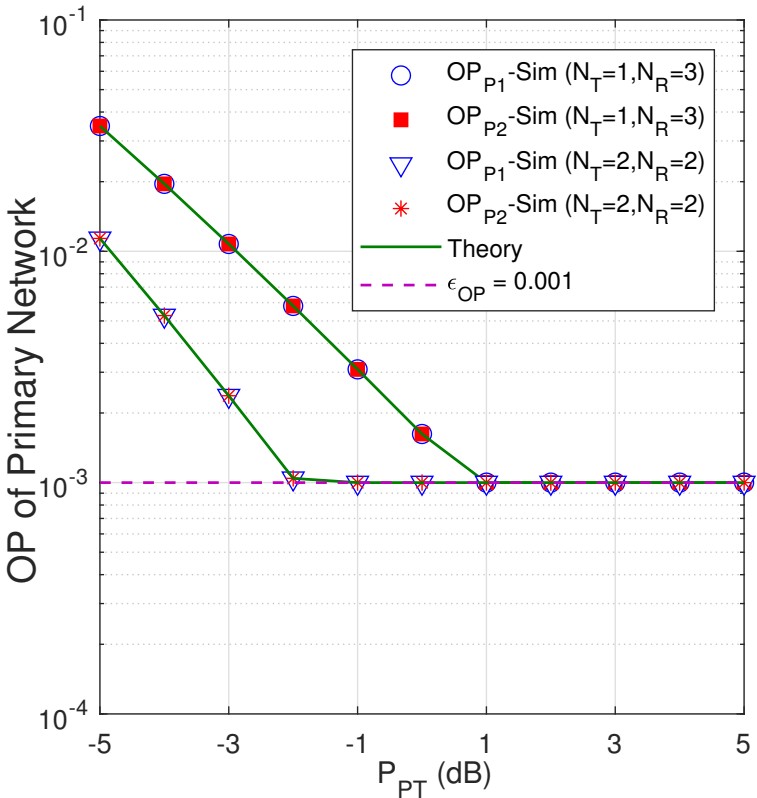

**Figure 2.** OP of the primary network as a function of $P_{PT}$ (dB) when $x_R = 0.3$.

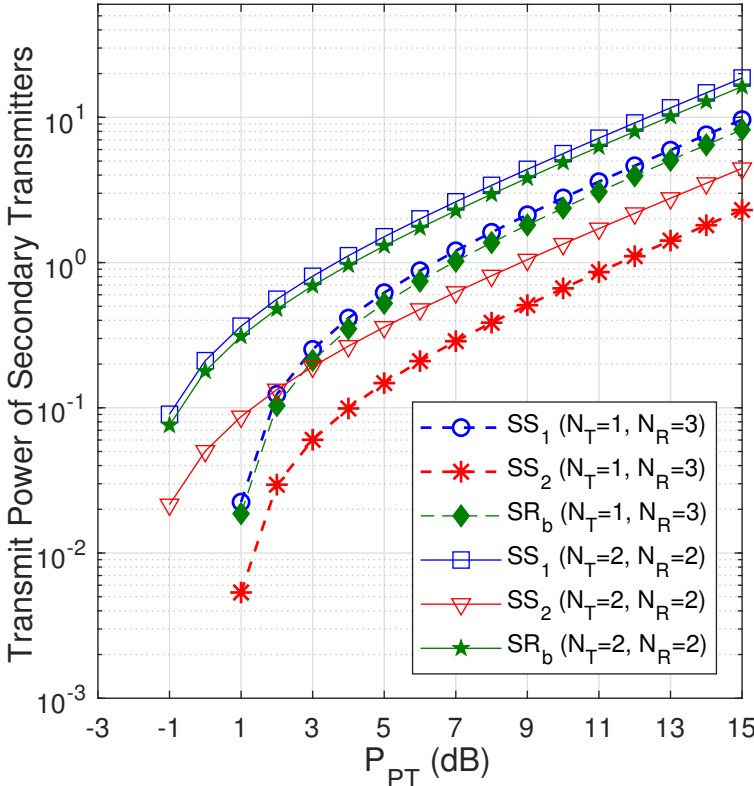

**Figure 3.** Transmit power of the secondary transmitters as a function of $P_{PT}$ (dB) when $x_R = 0.3$.

Now, we consider an optimization problem for the primary OP performance, where the total number of transmit and receive antennas is fixed by $N_T + N_R = N_{tot}$, and $N_{tot}$ is a constant. From (24), (25), and (27), $OP_{P1}$, $OP_{P2}$, and $OP_{P0}$ obtain the lowest value when

$$N_T = \begin{cases} \frac{N_{tot}}{2}, & \text{if } N_{tot} \text{ is even} \\ \frac{N_{tot}-1}{2} \text{ or } \frac{N_{tot}+1}{2}, & \text{if } N_{tot} \text{ is odd} \end{cases} \tag{56}$$

For example, in Figures 2 and 3, since $N_{tot} = 4$, the optimal values of $N_T$ and $N_R$ are $N_T = N_R = 2$.

### 4.2. OP of the Secondary Network

This sub-section studies the OP of the secondary network with $N_T = N_R = 2$. As presented in Figures 2 and 3, $P_{PT}$ should be higher than $-1$ (dB) so that the secondary transmitters are allowed to use the licensed band.

In Figure 4, we present the OP at the secondary sources as a function of $P_{PT}$ in dB with different values of the number of relays ($M$), and with $x_R = 0.3$. At first, we can see that the simulation results verify the correction of the expressions of $OP_{SS_1}$ and $OP_{SS_2}$ derived in Section 3. Then, we can see that $OP_{SS_2}$ is lower than $OP_{SS_1}$ because the datum $x_1$ has not yet been decoded by $SR_b$ at the first time slot. It is also seen from Figure 4 that the OP of $SS_1$ and $SS_2$ is lower with the increase in the number of relays ($M$). However, as observed in $OP_{SS_2}$, with $M = 3$ and $M = 8$, $OP_{SS_2}$ only changes slightly. Finally, we can observe that $OP_{SS_1}$ and $OP_{SS_2}$ decrease with the increase in $P_{PT}$. However, both $OP_{SS_1}$ and $OP_{SS_2}$ rapidly converged to saturation values as $P_{PT}$ is sufficiently high. Moreover, the saturation values do not depend on $P_{PT}$, which means that the diversity order is equal to zero. To explain the saturation points in Figure 4, we first rewrite the SINRs given in (10), (12), and (18) at a high $P_{PT}$ regime as follows:

$$\psi_{SR_b,x_1} \overset{P_{PT} \to +\infty}{\approx} \frac{P_{SS_1} g_{SS_1 SR_b}}{P_{SS_2} g_{SS_2 SR_b} + P_{PT} g_{PT_t SR_b}}, \psi_{SR_b,x_2} \overset{P_{PT} \to +\infty}{\approx} \frac{P_{SS_2} g_{SS_2 SR_b}}{P_{PT} g_{PT_t SR_b}}, \tag{57}$$

$$\varphi_{SS_1} \overset{P_{PT} \to +\infty}{\approx} \frac{P_{SR_b} g_{SR_b SS_1}}{P_{PT} g_{PT_w SS_1}}, \quad \varphi_{SS_2} \overset{P_{PT} \to +\infty}{\approx} \frac{P_{SR_b} g_{SR_b SS_2}}{P_{PT} g_{PT_w SS_2}}. \tag{58}$$

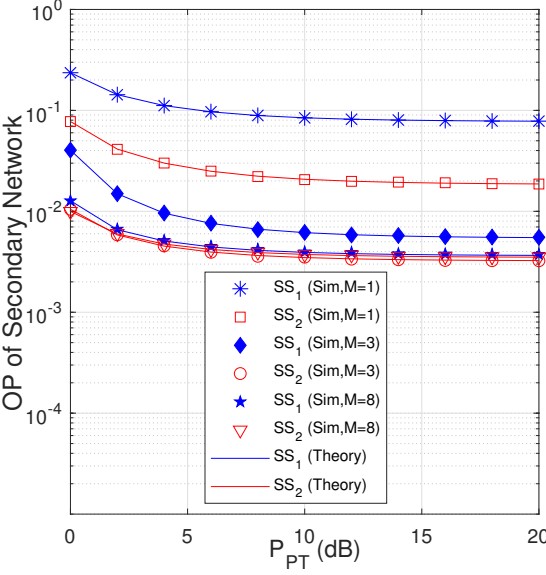

**Figure 4.** OP of the secondary sources versus $P_{PT}$ (dB) when $x_R = 0.3$.

From (35), (57), and (58), we can see that SINRs $\psi_{SR_b,x_1}$, $\psi_{SR_b,x_2}$, $\varphi_{SS_1}$, and $\varphi_{SS_2}$ at high $P_{PT}$ values do not depend on $P_{PT}$, and this is the reason why $OP_{SS_1}$ and $OP_{SS_2}$ converge to the saturation values.

Figure 5 presents $OP_{SS_1}$ and $OP_{SS_2}$ as functions of the number of relays ($M$) with different positions of the secondary relays and with $P_{PT} = 10$ (dB). As seen from Figure 5, $OP_{SS_1}$ and $OP_{SS_2}$ are lower with the increase in $M$. However, when $M$ is high enough, $OP_{SS_1}$ and $OP_{SS_2}$ converge to saturation values. For example, when $x_R = 0.4$, $OP_{SS_2}$ ($OP_{SS_1}$) converges to the constants as $M \geq 2$ ($M \geq 6$). Figure 5 also illustrates that, as $M$ increases, the performance gap between $OP_{SS_1}$ and $OP_{SS_2}$ is smaller. Therefore, an increasing $M$ also provides the performance fairness between $SS_1$ and $SS_2$. Finally, it can be observed that position of the secondary relays significantly affects $OP_{SS_1}$ and $OP_{SS_2}$. As seen from Figure 5, $OP_{SS_1}$ and $OP_{SS_2}$ with $x_R = 0.2$ are much lower than those with $x_R = 0.4$.

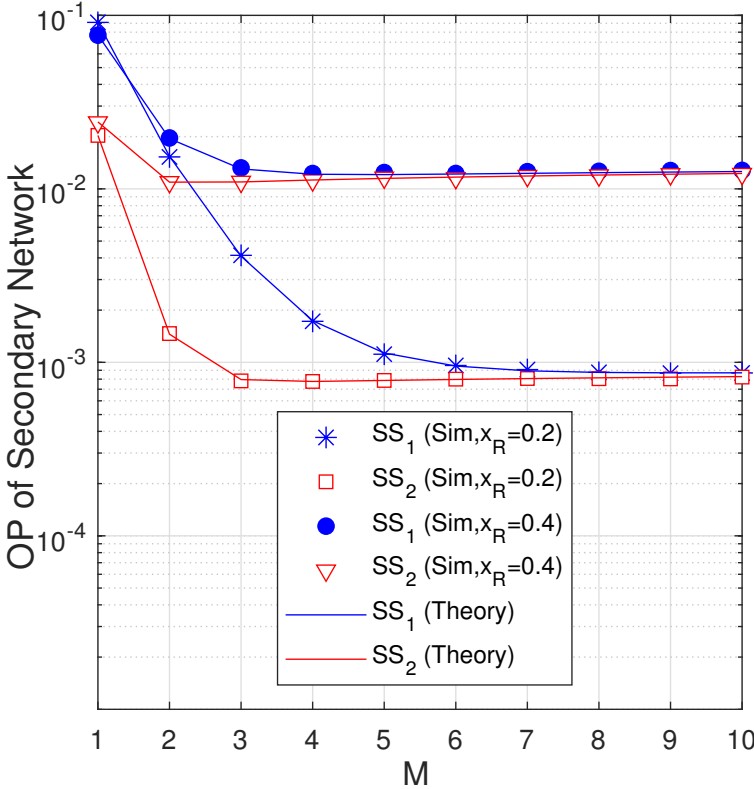

**Figure 5.** OP of the secondary sources as a function of $M$ when $P_{PT} = 10$ (dB).

To more clearly show the impact of $x_R$ on the OP performance of the secondary sources, in Figure 6, we present $OP_{SS_1}$ and $OP_{SS_2}$ as changing $x_R$ from 0.1 to 0.5. In this figure, $P_{PT}$ is fixed by 10 (dB). We first see that both $OP_{SS_1}$ and $OP_{SS_2}$ increase as $x_R$ increases, which means that the proposed scheme performs well when the secondary relays are near the source $SS_1$. Figure 6 also shows that $OP_{SS_1}$ and $OP_{SS_2}$ are lower with higher values of $M$, and the simulation results match very well with the analytical ones.

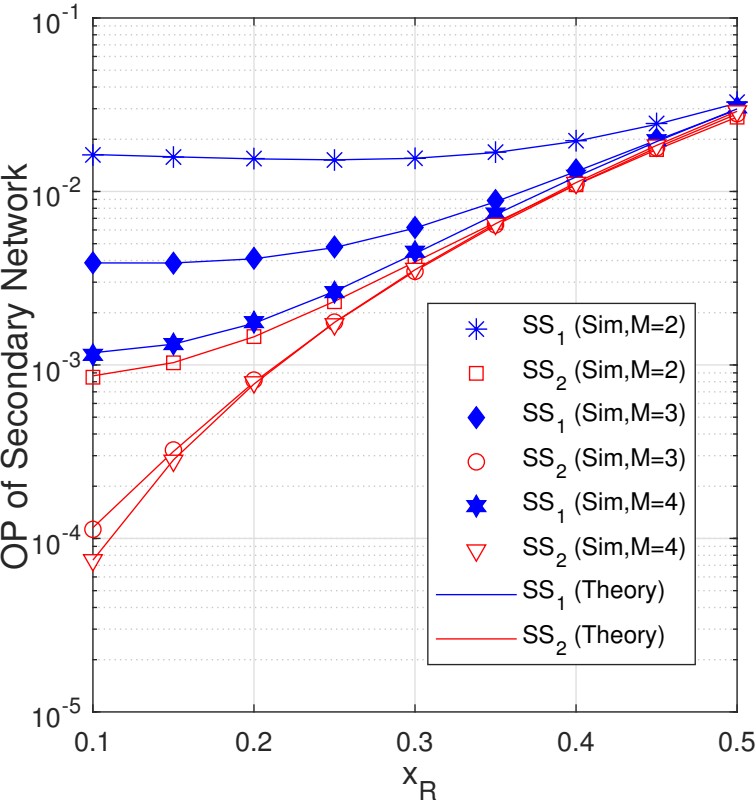

**Figure 6.** OP of the secondary sources as a function of $x_R$ when $P_{PT}$ = 10 (dB).

## 5. Conclusions

In this paper, we proposed the underlay CRN using TAS/SC to enhance the OP performance for the primary network, and using DNC TWR, ICT, and PRS to enhance the performance of the secondary network, in terms of OP and data rate. Additionally, we derived the closed-form expressions of the transmit power of the secondary transmitters and exact closed-form expressions of OP at the secondary sources over the Rayleigh fading channel. To check the correction of the derived formulas, Monte Carlo simulations were realized. Because all the derived expressions are in closed-form, they can be easily used to design and optimize the considered network. Then, as proven in Section 4, enhancing the primary OP performance with TAS/SC also increased the spectrum access possibility and transmit power of the secondary transmitters as well as increased the second OP performance. However, under the impact of the co-channel interference from the primary network, the secondary network only obtains the coding gain (i.e., the diversity gain equal to zero). The results show that the OP performance of the secondary network could be improved by increasing the number of secondary relays and placing the relays near one of the secondary sources. Finally, it is worth noting that increasing the number of secondary relays also provides performance fairness for the secondary sources.

**Author Contributions:** The main contributions of P.M.N., H.D.H., and P.V.T. were to review related works and conceive the main ideas; the main contributions of L.-T.T. and T.T.D. were to execute the performance evaluation and Monte Carlo simulations; the main contributions of T.T.D. and T.H. were to revise and finalize the manuscript. All authors have read and agreed to the published version of the manuscript.

**Funding:** This research received no external funding.

**Conflicts of Interest:** The authors declare no conflict of interest.

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
