# Peer review of "Outage Performance of Interference Cancellation-Aided Two-Way Relaying Cognitive Network with Primary TAS/SC Communication and Secondary Partial Relay Selection"

_electronics, doi:10.3390/electronics11223645_

Round 1

Reviewer 1 Report

This work designs a two-way relaying cognitive network and implements TAS/SC for primary communication and secondary partial relay selection. The work is interesting but has room for improvement.

My comments are listed below:

1. The literature review can be improved. It is unclear how the various schemes in [2]-[27] relate to the current work. The review text simply summarizes each work without proper grouping and analysis.

2. I am questioning the amount of technical contribution compared to [40], with only more relays and block fading channels.

3. Please clarify the channel knowledge availability assumptions.

4. Please highlight the analytical contributions in a Theorem/Lemma-with-proof style.

5. The OP floor in Fig. 2 is a bit sharp. Could you please explain it?

6. I wonder if there is a beamforming aspect considered with multiple antenna transceivers.

Author Response

Dear the Reviewer,

 We would like to thank the Reviewer very much for reviewing the manuscript.

We have performed the recommended revision in-line with the Reviewers' comments. After this revision, we hope that the clarity and presentation of the paper have been improved significantly. We hope that this revision will remove any of the shortcomings of the original submission.

Finally, we would like to send to Reviewer the cover letter which is attached with this email.

Sincerely Yours,

Pham Minh Nam, Ha Duy Hung, Lam-Thanh Tu, Pham Viet Tuan, Tran Trung Duy, Tan Hanh

Reviewer 2 Report

The authors mainly studied the outage performance of a two-way relaying scheme in an underlay cognitive radio network. It is an interesting topic. Meantime, I would like to point out the followings:

1.    The abstract is clear and well structured.

2.    The introduction started with good background knowledge, addressed the tackled scientific issue, sufficiently surveyed the related works from the literature, and finally listed the main motivations. However, the authors used redundant style to report related studies. I would highly recommend the authors to utilize diverse collection of reporting verbs and phrases as they can help readers stay tuned to the surveyed literature. (For instance, the verb propose/s/ed have appeared 23 times in the manuscript whereas other reporting verbs such as suggested, exhibited, depicted, … etc. have not been used even once).

Some language errors and phrasal connections should be corrected. Here are some examples:

- Line 22: so that quality of service (QoS) of the primary network is not harmful

- Line 20-23: However, SUs …………However, it is difficult…… ( These sentences are confusing). Please, revise them.

- Line 26: was

- Line 52: is

- Line 58: , then XORs them

- Line 72: this technique uses more one time slot than ANC TWR (should be: ‘ this technique uses one time slot more than what ANC TWR does’).

- Line 83-84: Although…….., however…..

- Line 85-86: between two the sources

- Line 88: did not consider…… (better to use neither….nor )

- Line 108: at two the sources

3.    Section 2 presented the system model. I find this section is good and well developed. Meanwhile, here are some issues that I would like the authors to respond to:

-       You mentioned that based on SC, the primary receiver (PR) selects one antenna out of NR ones to receive the signal. However, you did not clearly state how? Which measure will the SC rely on to pick that antenna? How?

-       Figure 1 clearly and adequately illustrates the proposed model. The only suggestion I have is to mark the multiple antennas at the transmitter and the receiver by (1st, 2nd, …, NT) and (1st, 2nd, …, NR), respectively. It will be more informative to do so instead of just NT and NR.

Some language errors:

-       Line 122: above (use “earlier”)

-       Line 132: close together (close to each other)

-       Line 133: for all the A and B nodes and for all the values of m

-       Line 135: (i.e., SS2). Is there something missing here?

-       Line 150: below

-       Line 150: can be removed it out

4.    The results section has sufficiently presented the main results and achievements. I would like to address the following:

-       Line 194: Next, we fix the some

-       Line 221: OP of both the sources

-       Line 221: decrease as increasing

-       Line 225: are lower as

-       Line 230: impacts on

5.    The conclusion part is good and includes the main ideas of the suggested scheme. However, it did not emphasis the main achievements in numbers or improvement percentage. It would be great to include them in this section.

Eventually, I encourage the authors to add a paragraph where they can catch the readers’ attention by clearly emphasizing the applications in which the presented scheme is applicable. How can this proposed model help improve the performance of these applications?

I would leave the authors with the right to choose the best place in the manuscript for such paragraph.     

Author Response

(The authors gave the same response as above.)

Round 2

Reviewer 1 Report

The paper is ok to be published.